# Prevalence and determinants of anxiety and stress in the general population during COVID-19 pandemic in Iraq: A cross-sectional study

**Banaz A. Saeed[1], Saween K. Ismael[2], Redar Muhmed Amin[3], Nasraw Mustafa Mahmud[1], Dara A. Al-Banna[4], Nazar P. Shabila[5,6]** *

**1** Department of Psychiatry, College of Medicine, Hawler Medical University, Erbil, Iraq, **2** Department of Educational and Psychological Counselling, College of Education, University of Salahaddin, Erbil, Kurdistan Region, Iraq, **3** Hawler Teaching Hospital, Erbil, Kurdistan Region, Iraq, **4** Department of Nursing, Faculty of Nursing, Tishk International University, Erbil, Iraq, **5** Department of Medical Laboratory Sciences, Catholic University in Erbil, Erbil, Kurdistan Region, Iraq, **6** Department of Community Medicine, College of Medicine, Hawler Medical University, Erbil, Kurdistan Region, Iraq

* nazarshabila@gmail.com

**Data Availability Statement:** All the data of this study are available from the Mendeley database at https://doi.org/10.17632/vx3k9md3jd.1.

## Abstract

### Background

The COVID-19 pandemic and its prevention and control measures, such as social distancing, self-isolation, and quarantine, have a negative impact on the population's mental health. This study aimed to determine the prevalence of anxiety and stress among the general population during the outbreak of COVID-19 and assess their associated factors.

### Methods

We carried out a cross-sectional study in Erbil governorate, Iraqi Kurdistan Region, from July 18 to September 12, 2020. We used an online survey questionnaire to collect data from a sample of Erbil population. The 10-items Perceived Stress Scale and the 7-item Generalized Anxiety Disorder scale were used to measure the levels of stress and anxiety.

### Results

A total of 851 persons responded to the survey. The prevalence of moderate and high perceived stress was 59.6% and 16.6%, respectively. The prevalence of moderate and severe anxiety was 24.7% and 22.7%, respectively. Age, gender, economic level, having contact with COVID-19 patients, and following COVID-19 news were independent variables significantly associated with stress levels. Age, gender, economic level, employment status, having symptoms of COVID-19, having contact with COVID-19 patients, and following COVID-19 news were independent variables significantly associated with anxiety levels.

**Funding:** The author(s) received no specific funding for this work.

**Competing interests:** The authors have declared that no competing interests exist.

## Conclusion

A high proportion of people experienced stress and anxiety during the COVID-19 outbreak in Erbil, Iraqi Kurdistan Region. Females, younger age, poor, and unemployed reported significantly higher stress and anxiety levels. There is a need to establish mechanisms at the population level to decrease the stress and anxiety risks and provide mental health coping measures in times of crisis, such as education about positive thinking, stress management programs, and the role of social support.

## Introduction

Infectious diseases are important risks to peoples' life and well-being and continue to be important causes of mortality and morbidity. New and reemerging infectious diseases are considered a continuous and important risk to the health and well-being of people in developing and developed countries [1].

The severe acute respiratory syndrome coronavirus 2 (SARS-CoV-2) infection outbreak began in Wuhan, China, in December 2019. The coronavirus disease 2019 (COVID-19) was identified as the cause of the disease. The World Health Organization soon considered the COVID-19 epidemic a rapidly growing pandemic throughout the world and a public health emergency [2].

The COVID-19 pandemic is a global health threat [3]. The primary mode of transmission of the virus is through person-to-person transmission by large airborne droplets. The virus is also transmitted through contact with contaminated surfaces [4].

Previous research has shown a deep and wide range impact of infection outbreaks on the mental health of people at the individual and community levels. People have experienced fear of getting ill or dying, feelings of helplessness, and stigma at an individual level [5]. At the initial stage of the COVID-19 pandemic, the general medical complications have received priority attention rather than the mental health effects. Later, research from different settings assessed the effect of the COVID-19 pandemic on different aspects of mental health in different segments of the population [6–10].

A study from Bangladesh has shown that a large portion of the general population reported mental health problems such as loneliness, depression, anxiety, and sleep disturbance during the COVID-19 pandemic in Bangladesh. Poor mental health was associated with female sex, unemployment, being a student, obesity, and living without a family [11]. Another study from Portugal showed that moderate to severe depression, anxiety, and stress were prevalent among the general population during the pandemic, with around half of the population reporting moderate or severe psychological impact of the outbreak. The study identified women, the unemployed, those with lower education, living in rural areas, and with flu-like symptoms or chronic disorders as the primary risk factors [10].

The COVID-19 pandemic also severely impacted university students' emotional well-being [12, 13], which was associated with marital status, financial condition, and education level [12]. Mental health issues were also reported among a high proportion of healthcare professionals during the COVID-19 pandemic. They were significantly associated with the working environment, economic condition, education level, area of residence, marital status, gender differences, professional category, body mass index, and smoking habit [9, 14].

The coronavirus outbreak can lead to different health problems, such as insomnia, anxiety, stress, depression, denial, fear, and anger [11, 15]. The COVID-19 pandemic and related public

health measures such as social distancing, self-isolation, and quarantine might negatively impact mental health. Loneliness and limited social interaction can be important risk factors for different mental health problems and even suicide [13, 16, 17]. The overall concerns might affect the daily behaviors, prevention, economy, and decision-making of health institutions, authorities, and policymakers that might weaken coronavirus disease control strategies, resulting in higher morbidity and more mental health needs at different levels [15, 16].

With the growing literature and evidence on the impact of the COVID-19 epidemic on people's psychological and mental health during the COVID-19 epidemic, limited research has examined the psychological impact of COVID-19 on the general public in Erbil. We conducted this study to see if the profile of mental health during the pandemic and the associated factors in Erbil are similar or different from other settings. More specifically, this study aimed to determine the prevalence of anxiety and stress among the general population during the outbreak of COVID-19 and assess their associated factor. This may enable public health authorities, healthcare managers, and providers in Erbil and similar settings to protect the community's psychological health from the outbreak of COVID-19 or similar outbreaks in the future.

## Materials and methods

### Design and setting

We carried out a cross-sectional study using an online survey questionnaire in Erbil, Iraqi Kurdistan Region, from July 18 to September 12, 2020. By September 12, 2020, 35983 COVID-19 cases and 1320 deaths were recorded in Iraqi Kurdistan Region, including 10467 cases and 486 deaths in Erbil governorate. The average number of COVID-19 cases per day was around 600 cases in the Iraqi Kurdistan Region and around 200 cases in Erbil governorate. After the survey, the number of COVID-19 cases and deaths continued to increase through several waves. By the end of September 2022, 465,827 COVID-19 cases and 7460 deaths were recorded in Iraqi Kurdistan Region, including 149,614 cases and 2185 deaths in Erbil governorate.

### Study participants

Around 1.6 million people live in Erbil governorate, including 878,000 persons in Erbil city. For the selection of the participants, the aim was to involve people from different social, educational, and economic statuses.

The sample size was calculated using the Epi-info based on a population size of 878,000 and assuming that the prevalence of stress among the general population in the COVID-19 pandemic context is 24.6% based on a previous study from neighboring Iran [18]. We found that a sample size of 791 persons was sufficient to achieve a 95% confidence interval for the prevalence with ±3% precision. The sample size was increased to 1000 to account for non-response.

People aged between 20 and 65 years who could understand and independently respond to the questions and were known to have access to the internet and social media were included in this survey. Exclusion criteria included people with pre-existing mental health problems or psychiatric disorders and having acute medical conditions during the survey.

### Sampling

It was not possible to obtain complete lists of Erbil population with contact details. Therefore, we prepared a list of potential participants from the surrounding communities in Erbil city by consulting with a number of key informants in the community to prepare these lists. The key informants included the community leader in some quarters (mukhtar), members of women's

associations, and food ration agents that usually have the contact information of the people in their respective areas or quarters. A simple random sample of 1000 persons was selected from the initial list of potential participants.

## Survey tool

We designed an online survey questionnaire based on Google Forms, which included three main parts. The first part of the questionnaire was about the demographic characteristics of the participants, such as age, sex, marital status, and occupation. It also included questions on hours spent at home and listening to corona news, history of infection symptoms during the last two weeks, history of contact with COVID-19 cases during the last two weeks, and history of comorbid medical problems. The second and third parts were about the assessment of stress and anxiety.

The second part of the survey questionnaire included the 10-items Perceived Stress Scale, which measured the stress level in the past month based on a 5 point Likert scale [19]. The Perceived Stress Scale has established acceptable psychometric properties. The scale is a commonly used, validated, and easy-to-use stress questionnaire [20]. The participants were asked to select the frequency of feeling or thinking a certain way over the last month from never to very often (0 = never, 1 = almost never, 2 = sometimes, 3 = fairly often, and 4 = very often). The total perceived stress scale score was calculated by summation of the scores of each question. Higher scores means having a greater stress level. The scores of the four positively stated items (items 4, 5, 7, and 8) were reversed (i.e., 0 = 4, 1 = 3, 2 = 2, 3 = 1, and 4 = 0). The total sum score of the perceived stress scale can range from 0 to 40. The interpretation of the Perceived Stress Scale total scores included low stress (0–13 scores), moderate stress (14–26 scores), and high stress (27–40 scores)

The third part of the survey questionnaire included the 7-item Generalized Anxiety Disorder (GAD-7) scale, which was used to measure anxiety levels [21]. The psychometric properties of this scale are acceptable [22, 23]. The participants responded to seven questions about anxiety over the past two weeks based on a 4 point Likert scale (0 = not at all, 1 = several days, 2 = over half a day, 3 = nearly every day. The GAD-7 total sum score ranges from 0 to 21. The interpretation of this measurement's total scores included normal (0–4 scores), mild anxiety (5–9 scores), moderate anxiety (10–14 scores), and severe anxiety (15–21 scores).

The survey questionnaire was translated into the Kurdish language. The translation was validated by back-translation to the English language by a native Kurdish speaker fluent in English. The validated questionnaire in the Kurdish language was pilot tested before administering it to the respondent.

The validity and applicability of the perceived stress component of the questionnaire showed an internal consistency (Cronbach's alpha) of 0.82 and a reliability coefficient of 0.70. The validity and applicability of the generalized anxiety disorder component of the questionnaire showed an internal consistency (Cronbach's alpha) of 0.80 and a reliability coefficient of 0.72 [24].

## Data collection

The anonymous online self-administered survey questionnaire was shared with the potential participants through social media tools such as Viber, Facebook Messenger, WhatsApp, and Twitter. Follow up messages with the link were sent to the participants with delayed or no response.

## Ethical considerations

The Research Ethics Committee of Hawler Medical University approved this study. The survey was anonymous, and data security was guaranteed. Written informed consent was received online before respondents answered survey questions. The participants were permitted to terminate the survey at any time they chose.

## Data analysis

The statistical package for the social sciences (SPSS, version 23) was used for data analysis. Values for the demographic characteristics of the participants and the prevalence of anxiety and stress were expressed as numbers and percentages. Anxiety and stress scores were expressed as means and standard deviations. Comparison of anxiety and stress mean scores with different variables was conducted by t-test, ANOVA, and linear regression tests. A p-value of $\leq 0.05$ was considered statistically significant. We created dummy variables for categorical variables first, then conducted a multivariate linear regression analysis between anxiety/stress and the associated variables.

## Results

A total of 851 persons responded to the survey. The sociodemographic and clinical characteristics of the respondents are shown in Table 1. Most respondents were female (70.2%), between 20 and 29 years old (50.8%), with a college education (72.6%), married (56.2%), employed (66.6%), living inside cities (87%), and from medium economic level (86.8%).

The average score on the perceived stress scale was 19.04 ± 7.59, indicating that the respondents had a moderate stress level. The average score on anxiety was 9.61 ± 5.69, indicating that the respondents had a mild level of anxiety. Of the 851 respondents, 203 (23.9%) had low perceived stress, 507 (59.6%) had moderate stress, and 141 (16.6%) had high stress, while 189 (22.2%) had no anxiety, 259 (30.4%) had mild anxiety, 210 (24.7%) had moderate anxiety, and 193 (22.7%) had severe anxiety. The prevalence of different degrees of stress and anxiety among the study participants is in Table 2.

Table 3 shows the association between sociodemographic and clinical factors with self perceived stress and anxiety. The mean perceived stress scale score was significantly higher in females than males (20.1 vs. 16.6, P <0.001), and, in those aged <40 years than the older respondents (19.5 vs. 17.1, P <0.001), in unemployed than employed (20.6 vs. 18.5, P = 0.015), and in poor than medium and very good economic levels (24.4 vs. 18.9 and 17.5, P <0.001). The mean perceived stress scale score was significantly higher in those who had COVID-19 symptoms in the last 14 days (P = 0.006), those who stayed home for >12 hours per day (P = 0.003), those with contact with COVID-19 patients (P <0.001), and those who frequently followed COVID-19 news (P <0.001).

The mean anxiety score was significantly higher in females than males (10.4 vs. 7.7, P <0.001), in those aged <40 years than the older respondents (9.9 vs. 8.3, P <0.001), in those with primary/secondary education and college education than postgraduate education (9.7 and 9.8 vs. 8.7, P <0.001), and in poor than medium and very good economic levels (13.2 vs. 9.5 and 8.5, P = 0.022). The mean anxiety score was significantly higher in those who had COVID-19 symptoms in the last 14 days (P = 0.003), those who stayed home for >12 hours per day (P = 0.005), those with contact with COVID-19 patients (P = 0.014), and those who frequently followed COVID-19 news (P <0.001).

We evaluated the effect of the respondents' sociodemographic and clinical characteristics on their stress levels through a linear regression model. All variables were included in the model. The age, gender, economic level, contact with COVID-19 patients, and following

**Table 1. Sociodemographic and clinical characteristics of the respondents.**

| Gender | No. | % |
|---|---|---|
| Male | 254 | 29.8 |
| Female | 597 | 70.2 |
| **Age (years)** | | |
| 20–29 | 432 | 50.8 |
| 30–39 | 255 | 30.0 |
| 40–49 | 120 | 14.1 |
| ≥50+ | 44 | 5.2 |
| **Education level** | | |
| Primary/secondary level | 108 | 12.7 |
| College level | 618 | 72.6 |
| Postgraduate level | 125 | 14.7 |
| **Marital status** | | |
| Single | 373 | 43.8 |
| Married | 478 | 56.2 |
| **Occupation** | | |
| Unemployed | 111 | 13.0 |
| Employed | 567 | 66.6 |
| Student | 173 | 20.3 |
| **Residence area** | | |
| Inside city | 740 | 87.0 |
| Outside city | 111 | 13.0 |
| **Economic level** | | |
| Poor | 37 | 4.3 |
| Medium | 739 | 86.8 |
| Very good | 75 | 8.8 |
| **Chronic disease** | | |
| No | 779 | 91.5 |
| Yes | 72 | 8.5 |
| **COVID-19 symptoms in the last 14 days** | | |
| No | 681 | 80.0 |
| Yes | 170 | 20.0 |
| **Hours staying at home per day** | | |
| <6 hours | 55 | 6.5 |
| 6–12 hours | 153 | 18.0 |
| >12 hours | 643 | 75.5 |
| **Contact with COVID-19 patient** | | |
| No | 574 | 67.5 |
| Yes | 277 | 32.5 |
| **Follow COVID-19 news** | | |
| None | 73 | 8.6 |
| Close to none | 54 | 6.3 |
| Sometimes | 516 | 60.6 |
| Many times | 208 | 24.4 |
| **Total** | **851** | **100.0** |

Table 2. Prevalence of different degrees of stress and anxiety among the participants.

| Condition | No. | % |
|---|---:|---:|
| **Stress** | | |
| Low | 203 | 23.9 |
| Moderate | 507 | 59.6 |
| High | 141 | 16.6 |
| **Anxiety** | | |
| No | 189 | 22.2 |
| Mild | 259 | 30.4 |
| Moderate | 210 | 24.7 |
| Severe | 193 | 22.7 |

COVID-19 news affected respondents' stress levels. The effect of these variables on the stress level included R = .357 and R2 = .128 (Table 4).

We evaluated the effect of the respondents' sociodemographic and clinical characteristics on their anxiety levels using a linear regression model. The age, gender, economic level, employment status, symptoms of COVID-19, contact with COVID-19 patients, and following COVID-19 news affected respondents' anxiety levels. The effect of these variables on the stress level included R = .338 and R2 = .1114 (Table 5).

## Discussion

This study assessed the stress and anxiety prevalence and severity in a sample of the general population in Erbil governorate during the COVID-19 outbreak. It also assessed the factors primarily associated with stress and anxiety. The prevalence of moderate and high stress was relatively high among the study sample. Another study showed a similarly high rate of severe stress (18%) among the adult population in the Iraqi Kurdistan Region [25]. The prevalence of moderate to severe stress reported in our study was higher than in other settings during the COVID-19 outbreak. A study from Iran showed a lower rate of moderate to extremely severe stress (24.6%) among the people [18]. A systematic review and meta-analysis from Iran showed that the prevalence of stress in five studies was 29.6% [26]. In Spain, 22.4% of people had moderate to extremely severe stress [27]. A lower prevalence of moderate to severe stress was reported in Portugal (5.6%) and China (8.1%) [10, 28]. The difference in the level of stress among the population in different settings might be related to cultural factors, the extent of the outbreak of COVID-19 and its severity during the study time, the study tools used to measure the stress level, and the methodological issues of different studies. The higher prevalence of moderate to severe stress reported in our study could be related to different factors. The type of response of the local government, health authorities, media, and social media to the outbreak plays a vital role in determining the stress level among the population of a specific country or region. Methodological limitations of this study can also be responsible for having a higher prevalence of stress. Including a higher number of female participants with a higher stress level might be responsible for a generally higher stress prevalence in this study.

The prevalence of moderate and severe anxiety was relatively high among the sample. Other studies from Iraq and the Iraqi Kurdistan Region reported a similar level of anxiety during the pandemic. For example, a study on the Iraqi Kurdistan adult population during the COVID-19 pandemic revealed that 47% had anxiety [25]. Another study from Iraq showed that 49.5% of people in the whole country and 45.1% in the northern governorates had anxiety [29]. The prevalence of moderate to severe anxiety reported in our study was higher than in

**Table 3. Factors associated with self-perceived stress and anxiety in the sample.**

| Variable | N | Stress | | | | | Anxiety | | | | |
|---|---|---|---|---|---|---|---|---|---|---|---|
| | | Mean | SD | t/F | df | P value | Mean | SD | t/F | df | P value |
| **Gender** | | | | | | | | | | | |
| Male | 254 | 16.6 | 7.2 | -6.164 | 849 | <0.001 | 7.7 | 5.4 | -6.436 | 849 | <0.001 |
| Female | 597 | 20.1 | 7.5 | | | | 10.4 | 5.6 | | | |
| **Age** | | | | | | | | | | | |
| <40 | 687 | 19.5 | 7.5 | 3.754 | 849 | <0.001 | 9.9 | 5.7 | 3.316 | 849 | 0.001 |
| ≥40 | 164 | 17.1 | 7.8 | | | | 8.3 | 5.5 | | | |
| **Education level** | | | | | | | | | | | |
| Primary/secondary level | 108 | 18.6 | 8.8 | 0.723 | 849 | 0.485 | 9.7 | 6.4 | 1.868 | 849 | <0.001 |
| College level | 618 | 19.2 | 7.5 | | | | 9.8 | 5.6 | | | |
| Postgraduate level | 125 | 18.5 | 6.7 | | | | 8.7 | 5.3 | | | |
| **Marital status** | | | | | | | | | | | |
| Single | 373 | 19.5 | 7.7 | 1.451 | 849 | 0.147 | 9.7 | 5.7 | 0.364 | 849 | 0.716 |
| Married | 478 | 18.7 | 7.5 | | | | 9.5 | 5.7 | | | |
| **Occupation** | | | | | | | | | | | |
| Unemployed | 111 | 20.6 | 7.9 | 4.245 | 849 | 0.015 | 11.4 | 5.5 | 8.602 | 849 | 0.155 |
| Employed | 567 | 18.5 | 7.7 | | | | 9.1 | 5.7 | | | |
| Student | 173 | 19.7 | 6.8 | | | | 10.1 | 5.4 | | | |
| **Residence area** | | | | | | | | | | | |
| Inside city | 740 | 19.2 | 7.6 | 1.449 | 849 | 0.148 | 9.6 | 5.7 | 0.368 | 849 | 0.713 |
| Outside city | 111 | 18.1 | 7.4 | | | | 9.4 | 5.9 | | | |
| **Economic level** | | | | | | | | | | | |
| Poor | 37 | 24.4 | 6.2 | 11.261 | 849 | <0.001 | 13.2 | 5.9 | 9.134 | 849 | 0.022 |
| Medium | 739 | 18.9 | 7.4 | | | | 9.5 | 5.6 | | | |
| Very good | 75 | 17.5 | 8.7 | | | | 8.5 | 5.7 | | | |
| **Chronic disease** | | | | | | | | | | | |
| No | 779 | 19.0 | 7.6 | -0.389 | 849 | 0.700 | 9.6 | 5.7 | -0.285 | 849 | 0.776 |
| Yes | 72 | 19.4 | 7.9 | | | | 9.8 | 6.0 | | | |
| **Symptoms in the last 14 days** | | | | | | | | | | | |
| No | 681 | 18.7 | 7.4 | -2.760 | 849 | 0.006 | 9.3 | 5.6 | -2.975 | 849 | 0.003 |
| Yes | 170 | 20.5 | 8.0 | | | | 10.8 | 6.0 | | | |
| **Stay home** | | | | | | | | | | | |
| ≤12 hours | 208 | 17.7 | 7.2 | -2.970 | 849 | 0.003 | 8.6 | 5.4 | -2.810 | 849 | 0.005 |
| >12 hours | 643 | 19.5 | 7.7 | | | | 9.9 | 5.8 | | | |
| **Contact with COVID-19 patients** | | | | | | | | | | | |
| No | 574 | 18.4 | 7.5 | -3.627 | 849 | <0.001 | 9.3 | 5.6 | -2.455 | 849 | 0.014 |
| Yes | 277 | 20.4 | 7.7 | | | | 10.3 | 5.7 | | | |
| **Following COVID-19 news** | | | | | | | | | | | |
| None/almost none | 127 | 16.8 | 8.5 | 7.868 | 849 | <0.001 | 8.6 | 6.1 | 3.828 | 849 | <0.001 |
| Sometimes | 516 | 19.2 | 7.4 | | | | 9.6 | 5.6 | | | |
| A lot | 208 | 20.1 | 7.2 | | | | 10.3 | 5.6 | | | |

SD: standard deviation, t: t-test value, F: F test value, df: degree of freedom

other settings during the COVID-19 outbreak. A systematic review and meta-analysis from Iran showed that the prevalence of anxiety in 17 studies was 31.9% [26]. The prevalence of moderate to extremely severe anxiety was 31.8% in Iran [18] and 34% in Bangladesh [11]. In a

**Table 4. Analysis of sociodemographic and clinical characteristics of the respondents on stress level by regression analysis.**

| Characteristics | Unstandardized Coefficients | | Standardized Coefficients | t | P value |
|---|---|---|---|---|---|
| | B | Std. Error | Beta | | |
| Constant | 19.649 | 1.700 | | 11.558 | <0.001 |
| Gender (female) | 3.364 | 0.601 | 0.203 | 5.599 | <0.001 |
| Age (≥40 years) | -2.348 | 0.712 | -0.122 | -3.297 | 0.001 |
| Education level (college) | -0.352 | 0.779 | -0.021 | -0.452 | 0.651 |
| Education level (postgraduate) | -0.043 | 1.016 | -0.002 | -0.043 | 0.966 |
| Marital status (married) | -0.346 | 0.579 | -0.023 | -0.598 | 0.550 |
| Occupation (employed) | -0.636 | 0.795 | -0.040 | -0.799 | 0.424 |
| Occupation (student) | 0.062 | 0.924 | 0.003 | 0.067 | 0.947 |
| Residence area (outside city) | -0.600 | 0.759 | -0.027 | -0.790 | 0.429 |
| Economic level (medium) | -4.786 | 1.231 | -0.213 | -3.886 | <0.001 |
| Economic level (very good) | -7.055 | 1.480 | -0.264 | -4.766 | <0.001 |
| Chronic disease (yes) | 0.670 | 0.930 | 0.025 | 0.721 | 0.471 |
| COVID-19 symptoms in the last 14 days (yes) | 0.995 | 0.638 | 0.052 | 1.559 | 0.119 |
| Stay home (≥12 hours) | 0.294 | 0.621 | 0.017 | 0.474 | 0.636 |
| Contact with COVID-19 patients (yes) | 1.997 | 0.549 | 0.123 | 3.636 | <0.001 |
| Following COVID-19 news (sometimes) | 2.124 | 0.719 | 0.137 | 2.957 | 0.003 |
| Following COVID-19 news (many times) | 3.255 | 0.828 | 0.184 | 3.932 | <0.001 |

study from Spain, 25.3% of people showed moderate to extremely severe anxiety [27]. In China, 28.8% of people reported moderate to severe anxiety symptoms [28]. The higher prevalence of anxiety reported in our study could be related to including a larger number of female participants who already had higher anxiety. The difference in the level of anxiety among the population in different settings might be related to cultural factors, the extent of the outbreak

**Table 5. Analysis of sociodemographic and clinical characteristics of the respondents on anxiety level by regression analysis.**

| Characteristics | Unstandardized Coefficients | | Standardized Coefficients | t | P value |
|---|---|---|---|---|---|
| | B | Std. Error | Beta | | |
| Constant | 10.618 | 1.285 | | 8.266 | <0.001 |
| Gender (female) | 2.710 | 0.454 | 0.218 | 5.970 | <0.001 |
| Age (≥40 years) | -1.378 | 0.538 | -0.096 | -2.561 | 0.011 |
| Education level (college) | -0.509 | 0.589 | -0.040 | -0.864 | 0.388 |
| Education level (postgraduate) | -0.736 | 0.767 | -0.046 | -0.958 | 0.338 |
| Marital status (married) | 0.421 | 0.438 | 0.037 | 0.961 | 0.337 |
| Occupation (employed) | -1.256 | 0.601 | -0.104 | -2.090 | 0.037 |
| Occupation (student) | -0.273 | 0.698 | -0.019 | -0.391 | 0.696 |
| Residence area (outside city) | 0.131 | 0.574 | 0.008 | 0.229 | 0.819 |
| Economic level (medium) | -3.023 | 0.930 | -0.180 | -3.249 | 0.001 |
| Economic level (very good) | -4.611 | 1.119 | -0.230 | -4.122 | <0.001 |
| Chronic disease (yes) | 0.278 | 0.703 | 0.014 | 0.396 | 0.692 |
| COVID-19 symptoms in the last 14 days (yes) | 0.957 | 0.482 | 0.067 | 1.985 | 0.047 |
| Stay home (≥12 hours) | 0.041 | 0.469 | 0.003 | 0.087 | 0.931 |
| Contact with COVID-19 patients (yes) | 1.036 | 0.415 | 0.085 | 2.495 | 0.013 |
| Following COVID-19 news (sometimes) | 0.791 | 0.543 | 0.068 | 1.458 | 0.145 |
| Following COVID-19 news (A lot) | 1.800 | 0.625 | 0.136 | 2.878 | 0.004 |

of COVID-19 and its severity during the study time, and the study tools used to measure the stress and anxiety level.

Previous outbreaks of other infectious diseases had also resulted in high mental and psychological symptoms. A study from the acute SARS outbreak showed that the people indirectly exposed to SARS in Taiwan experienced psychological symptoms during the outbreak. These symptoms were attributed to the impact of SARS, poor health conditions, lack of social support, and economic decline [30]. The psychological response of people to an infectious disease outbreak is complicated. Stress and anxiety can be caused by different issues such as the feeling of vulnerability, loss of control, health concerns, the transmissibility of infection, concern about family's health, changes in work conditions, economic difficulties, and isolation [31]. COVID-19 might increase personal risk perception since it is a highly transmissible infection [32, 33] and has high morbidity and mortality rates [34]. Moreover, compared to other emergency events, the people affected by the SARS or COVID-19 might experience isolation more severely because of the stigma related to the disease and hostility from the public [30, 35]. Research has suggested that the stigma related to COVID-19 is an important source of mental distress, including anxiety, stress, and depression, particularly among the affected individuals and their families [35].

In this study, the female gender was an independent variable significantly associated with stress and anxiety. Female sex was also a significant independent factor associated with stress and anxiety in another study from the Kurdistan Region [25]. In Spain, a significantly higher proportion of females had stress and anxiety than males [27]. In contrast, the male gender was significantly associated with higher stress and anxiety scores in China [28]. In general, women are at higher risk of depression, anxiety, and stress, according to previous extensive epidemiological studies [36]. Several biological, psychological, and sociological mechanisms might contribute to the higher vulnerability of women, including the effect of sex hormones, females' reliance on emotion-focused coping styles, and gender differences in trauma type, symptom reporting, social support, and social roles [37].

In this study, younger age was an independent variable significantly associated with stress and anxiety. A similar finding was reported in Spain [27]. In another study from the Iraqi Kurdistan Region, age was not significantly associated with stress and anxiety [25]. Previous research has indicated that the elderly are primarily susceptible to the negative psychological consequences of critical and disaster situations [38]. However, similar to our findings, some studies on COVID-19-related mental health have found that age is considered a protective factor. Older victims might show less stress and anxiety symptoms due to their longer life experiences, previous disaster exposure experience, or fewer life responsibilities [39]. Research has found that loneliness and financial distress in younger adults are associated with poorer mental health outcomes. On the other hand, older adults show more resilience than young age groups, which may play a vital role in protection against mental health problems [40].

In this study, poor economic situation and unemployment were independent variables significantly associated with stress and anxiety. Similar findings were reported from Spain [27]. In another study from the Iraqi Kurdistan Region, the economic situation had no significant effect on stress and anxiety. However, unemployment was significantly associated with higher anxiety and lower stress than non-health-related occupations [25]. Research has shown that individuals are not only worried about health-related aspects during the COVID-19 pandemic, but they are also concerned about their economic vulnerability and their exposure to a negative economic recession [41].

In the current study, having contact with COVID-19 patients was a significant independent factor associated with stress and anxiety, while having COVID-19 symptoms in the last 14 days was a significant independent factor associated with anxiety only. In Spain, individuals

with close contact with COVID-19 patients and those with COVID-19 symptoms revealed significantly higher stress and anxiety than those without, but with a small effect [27]. Those in contact with COVID-19 patients can be anxious about getting the disease, its consequences, and transmitting the infection to their family members.

Following COVID-19 news was a significant independent factor associated with stress and anxiety. Research has demonstrated that individuals who frequently follow COVID-19 news experience higher psychological distress [42]. The news published on COVID-19 is always distressing, with some news containing rumors. Therefore, anxiety might increase when an individual continuously follows COVID-19 news [43]. Moreover, fabricated reports and misinformation on COVID-19 might worsen depressive symptoms in the population [26, 44].

This study has a number of limitations. The study population was limited to Erbil governorate, which limits the generalization of the results to other areas of Kurdistan and Iraq that might have been affected by the outbreak to a lesser or more extent. This study was conducted at the expanding stages of the outbreak as the number of patients and deaths rapidly increased. The level of stress and anxiety might have been different between the early stage of the pandemic and the later stages. In the early stage, there were many uncertainties surrounding the disease, with a higher possibility of stress and anxiety. In the later stages, many people were affected by COVID-19, recovered, and might have less stress and anxiety. With a cross-sectional study, it is difficult to attribute the high prevalence of anxiety and stress to the pandemic since there is no comparison with the pre-pandemic situation. The economic difficulties and political instability in Iraqi Kurdistan Region prior and during the study period are also stressors and can affect the anxiety and stress levels of the people. Around 70% of the respondents were females, which could be related to selection or random effect and a higher response rate or interest in the survey from the female sample. This could have potential consequences on the results towards a higher prevalence as females showed a significantly higher prevalence of stress and anxiety.

The use of online questionnaire is also associated with several limitations. As the response was anonymous, we cannot confirm that the intended respondents answered the questionnaire and that the survey link was not shared with and answered by others. As the questionnaire was sent through social media, people who did not have access to the internet and social media were possibly excluded, biasing the sample. Therefore, the respondents are mostly young and educated people. The older people and the less educated are a minority of the sample. Research has shown that the average response rate to online surveys is around 44% [45]. It is documented that sending an online survey to more participants will not result in a higher response rate. Therefore, sending online surveys to a clearly defined and refined population is recommended. The high response rate in our study (851 out of 1000) could be related to choosing a specifically targeted sample known to have access to the internet and social media. We also followed up with the invited participants by sending follow-up messages and resending the survey link.

## Conclusion

A considerably high proportion of people suffered from stress and anxiety during the COVID-19 outbreak in Erbil, Iraqi Kurdistan Region. Females, younger age, poor, and unemployed reported significantly higher stress and anxiety levels. There is a need to establish mechanisms at the population level to decrease the stress and anxiety risks and provide mental health coping measures in times of crisis. Potential interventions might include psychoeducation about the features of stress, how to deal with stress, practicing stress management programs, and educating people about positive thinking and the role of social support during the pandemic.

It is also important to provide psychoeducation to people about anxiety symptoms and their treatment if symptoms persist and do not improve. Additional research is required to assess stress and anxiety among people, better understand other factors associated with mental health symptoms, and determine and evaluate effective coping strategies.

## Author Contributions

**Conceptualization:** Banaz A. Saeed, Redar Muhmed Amin, Nasraw Mustafa Mahmud, Nazar P. Shabila.

**Data curation:** Banaz A. Saeed, Saween K. Ismael, Redar Muhmed Amin, Nasraw Mustafa Mahmud.

**Formal analysis:** Banaz A. Saeed, Dara A. Al-Banna, Nazar P. Shabila.

**Investigation:** Saween K. Ismael.

**Methodology:** Saween K. Ismael, Redar Muhmed Amin, Nasraw Mustafa Mahmud, Dara A. Al-Banna, Nazar P. Shabila.

**Validation:** Dara A. Al-Banna.

**Writing – original draft:** Banaz A. Saeed, Nazar P. Shabila.

**Writing – review & editing:** Saween K. Ismael, Redar Muhmed Amin, Nasraw Mustafa Mahmud, Dara A. Al-Banna.

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
