## [Decision Letter · Decision Letter 0]

8 Dec 2022

PONE-D-22-24670Prevalence and determinants of anxiety and stress in the general population during COVID-19 pandemicPLOS ONE

Dear Dr. Nazar P. Shabila,

Thank you for submitting your manuscript to PLOS ONE. After careful consideration, we feel that it has merit but does not fully meet PLOS ONE’s publication criteria as it currently stands. Therefore, we invite you to submit a revised version of the manuscript that addresses the points raised during the review process.

We look forward to receiving your revised manuscript.

Kind regards,

Md. Kamrul Hasan, MPH

Academic Editor

PLOS ONE

Journal Requirements:

Reviewers' comments:

Reviewer's Responses to Questions

**Comments to the Author**

1. Is the manuscript technically sound, and do the data support the conclusions?

Reviewer #1: Partly

Reviewer #2: Partly

Reviewer #3: Yes

2. Has the statistical analysis been performed appropriately and rigorously? 

Reviewer #1: Yes

Reviewer #2: Yes

Reviewer #3: Yes

3. Have the authors made all data underlying the findings in their manuscript fully available?

Reviewer #1: Yes

Reviewer #2: No

Reviewer #3: Yes

4. Is the manuscript presented in an intelligible fashion and written in standard English?

Reviewer #1: Yes

Reviewer #2: Yes

Reviewer #3: Yes

5. Review Comments to the Author

Reviewer #1: Dear Author,

This article entitled Prevalence and determinants of anxiety and stress in the general population during COVID-19 pandemic is interesting. I have read the paper and found it as a potential article for consideration for publication. However, there are some specific points that should be corrected before taken such decision.

1. The title should contain study type and place of the study, for example; Prevalence and determinants of anxiety and stress in the general population during COVID-19 pandemic in Iraq: A cross-sectional study

2. Average scores of mental health status presented in abstract do not make any sense rather the authors are suggested to present them as their significance/indications whether these values tell us about the mental health condition of participants.

3. The presented data are more than two years old and what about the current mental health status in Iraq that was supposed to disturbed by the pandemic. Because perception/experience about Covid-19 and associated matters are quickly evolving, therefore, the authors are asked to present similar findings from others studies in Iraq/similar demographics by detail literature review. This discussion/information would justify the need for publication of their own data. For these discussion the authors are suggested to discuss and cite the following articles in their article, but not limited to-

- Das R, Hasan MR, Daria S, Islam MR. Impact of COVID-19 pandemic on mental health among general Bangladeshi population: a cross-sectional study. BMJ Open. 2021;11(4):e045727. Published 2021 Apr 9. doi:10.1136/bmjopen-2020-045727

-Islam MR, Daria S, Das R, Hasan MR. A nationwide dataset on the mental health of the Bangladeshi population due to the COVID-19 pandemic. Data Brief. 2021;38:107347. doi:10.1016/j.dib.2021.107347

- Islam MR, Qusar MMAS, Islam MS. Mental Health of Children Amid COVID-19 Pandemic in Bangladesh: An Exploratory Observation. Asia Pac J Public Health. 2021;33(4):469-470. doi:10.1177/10105395211004371

-Nahar Z, Sohan M, Supti KF, et al. Prevalence and associated risk factors for mental health problems among female university students during COVID-19 pandemic: A cross-sectional study findings from Dhaka, Bangladesh. Heliyon. 2022;8(10):e10890. doi:10.1016/j.heliyon.2022.e10890

- Islam MR, Nahar Z, Hossain MS, et al. Prevalence and associated factors for elevated fear and depressive symptoms among the private service holders in Bangladesh during the Covid-19 pandemic: A cross-sectional study. Health Sci Rep. 2022;5(5):e795. Published 2022 Aug 23. doi:10.1002/hsr2.795

- Daria S, Islam MR. Increased suicidal behaviors among students during COVID-19 lockdowns: A concern of student's mental health in Bangladesh. J Affect Disord Rep. 2022;8:100320. doi:10.1016/j.jadr.2022.100320

- Repon MAU, Pakhe SA, Quaiyum S, Das R, Daria S, Islam MR. Effect of COVID-19 pandemic on mental health among Bangladeshi healthcare professionals: A cross-sectional study. Sci Prog. 2021;104(2):368504211026409. doi:10.1177/00368504211026409

-Islam MR, Quaiyum S, Pakhe SA, Repon MAU, Bhuiyan MA. Dataset concerning the mental health of healthcare professionals during COVID-19 pandemic in Bangladesh. Data Brief. 2021;39:107506. doi:10.1016/j.dib.2021.107506

4. The epidemiological data presented in this paper are also very old, they should be updated to let the readers know about the up-to-date information.

5. Getting 851 responses by inviting 1000 potential respondents are extremely rate in the context of biasness free online survey. As the global response rate is approximately 25% for all online survey. How the authors eliminate bianess of their study. How they selected participants and what was the assumtions befre data collection, these explanation and information should be given in the method section.

6. Why the autors took 70% female participants??? Was it randon effect, pls give explanation and potetianl consequences on the results.

Reviewer #2: I appreciate the opportunity to review the manuscript entitled "Prevalence and determinants of anxiety and stress in the general population during COVID-19 pandemic”. The study aimed to determine the prevalence of anxiety and stress among the general population during the outbreak of COVID-19 and assess their associated factors. This study is relevant and provides a significant contribution to the knowledge about the psychological impact of the 1st wave of COVID. However, there are some issues that must be addressed prior to publication. I have included my comments and recommendations below:

Introduction

P. 3, ln 61: The authors state: “limited research has addressed the direct effect of COVID-19 on mental health” – nowadays, there are a lot of research on this matter, so this information must be updated. The authors should refer to existing information, as well as the profile of people with more risk factors in terms of their mental health, in order to see if, in the area in question, the profile is similar or different. I think the relevance will be more in this sense, because there are already many studies.

Method

Study Participants

Additional information is needed about the key informants and the participants criteria that based the initial selection od the 1000 people.

If the questionnaire was sent through social media, is it possible that people without that kind of access were excluded, biasing the sample? As can be seen in the description of the participants, it is mostly young people who responded, which makes sense considering the way the questionnaire was disseminated. The older people are a minority of the sample, which should be mentioned in the limitations of the study.

Measures

Reliability information must be presented, for both scales.

Results

If the authors use scales like the Perceived Stress Scale and the Anxiety Scale, it makes no sense to present the statistics for each item, only the overall stress score.

The statistical information presented, such as t-test results, should be presented in greater detail, not only the p-values

Discussion

In the discussion there is no point in re-presenting statistical values.

How can the authors guarantee that the values found are due to the pandemic impact and do not reflect pre-pandemic values?

This should also be mentioned in the limitations, since this is a cross-sectional study

Reviewer #3: This is a well-written manuscript that highlights a topical issue of interest to global health. However, the paper can be improved by considering the under-listed comments.

1. Line 90: What is the total size of the population? How did you arrive at the sample size used?

2. Line 91: The inclusion and exclusion criteria should be clearly stated.

3. Lines: 92 to 96: The statements there should be moved to a section on sampling.

4. Line 96 to 98: From the sentence beginning “The online …. Should be moved to a section on data collection.

5. The authors used an online questionnaire to gather the data. What shows that the intended respondents answered the questionnaire? The limitations of the study should be stated including the weaknesses of using the approach to data collection employed.

6. The discussion section is highly descriptive. The authors can improve upon it by providing explanations as to why they reported a particular result and also explain why a particular study support or does not support their study.

7. The paper requires minor editing.

6. PLOS authors have the option to publish the peer review history of their article (what does this mean?). If published, this will include your full peer review and any attached files.

Reviewer #1: **Yes: **Md. Rabiul Islam

Reviewer #2: **Yes: **Rute Brites

Reviewer #3: No

---

## [Author Response · Author response to Decision Letter 0]

4 Jan 2023

Thank you very much for the valuable and useful comments and suggestions on our manuscript. They helped us to improve the quality and clarity of the manuscript significantly.

We have made the necessary revision by responding to the suggested comments. Please find below explanations to the revision made through a point-to-point response to the comments. All changes are highlighted.

Editor comment

Author's response

PLOS ONE style requirements are applied to the manuscript.

Author's response

The repository data of our study is now available and the DOI for the data repository is provided:

https://doi.org/10.17632/vx3k9md3jd.2

3. Please provide additional details regarding participant consent. In the Methods section, please ensure that you have specified (1) whether consent was informed and (2) what type you obtained (for instance, written or verbal). If your study included minors, state whether you obtained consent from parents or guardians. If the need for consent was waived by the ethics committee, please include this information.

Author's response

Details of receiving written informed consent are provided now (Page 8, lines 173-174). 

Reviewer #1: 

Reviewer’s comment 

This article entitled Prevalence and determinants of anxiety and stress in the general population during COVID-19 pandemic is interesting. I have read the paper and found it as a potential article for consideration for publication. However, there are some specific points that should be corrected before taken such decision.

Author's response

Thank you very much for the valuable comments.

Reviewer’s comment 

1. The title should contain study type and place of the study, for example; Prevalence and determinants of anxiety and stress in the general population during COVID-19 pandemic in Iraq: A cross-sectional study

Author's response

The title has been changed as suggested (Page 1, lines 1-2).

Reviewer’s comment 

2. Average scores of mental health status presented in abstract do not make any sense rather the authors are suggested to present them as their significance/indications whether these values tell us about the mental health condition of participants.

Author's response

The average score of mental health status are now removed from the Abstract and only the relevant prevalence of mental health conditions and their significance indications are provided (Page 2, lines 30).

Reviewer’s comment 

3. The presented data are more than two years old and what about the current mental health status in Iraq that was supposed to disturbed by the pandemic. Because perception/experience about Covid-19 and associated matters are quickly evolving, therefore, the authors are asked to present similar findings from others studies in Iraq/similar demographics by detail literature review. This discussion/information would justify the need for publication of their own data. For these discussion the authors are suggested to discuss and cite the following articles in their article, but not limited to-

- Das R, Hasan MR, Daria S, Islam MR. Impact of COVID-19 pandemic on mental health among general Bangladeshi population: a cross-sectional study. BMJ Open. 2021;11(4):e045727. Published 2021 Apr 9. doi:10.1136/bmjopen-2020-045727

-Islam MR, Daria S, Das R, Hasan MR. A nationwide dataset on the mental health of the Bangladeshi population due to the COVID-19 pandemic. Data Brief. 2021;38:107347. doi:10.1016/j.dib.2021.107347

- Islam MR, Qusar MMAS, Islam MS. Mental Health of Children Amid COVID-19 Pandemic in Bangladesh: An Exploratory Observation. Asia Pac J Public Health. 2021;33(4):469-470. doi:10.1177/10105395211004371

-Nahar Z, Sohan M, Supti KF, et al. Prevalence and associated risk factors for mental health problems among female university students during COVID-19 pandemic: A cross-sectional study findings from Dhaka, Bangladesh. Heliyon. 2022;8(10):e10890. doi:10.1016/j.heliyon.2022.e10890

- Islam MR, Nahar Z, Hossain MS, et al. Prevalence and associated factors for elevated fear and depressive symptoms among the private service holders in Bangladesh during the Covid-19 pandemic: A cross-sectional study. Health Sci Rep. 2022;5(5):e795. Published 2022 Aug 23. doi:10.1002/hsr2.795

- Daria S, Islam MR. Increased suicidal behaviors among students during COVID-19 lockdowns: A concern of student's mental health in Bangladesh. J Affect Disord Rep. 2022;8:100320. doi:10.1016/j.jadr.2022.100320

- Repon MAU, Pakhe SA, Quaiyum S, Das R, Daria S, Islam MR. Effect of COVID-19 pandemic on mental health among Bangladeshi healthcare professionals: A cross-sectional study. Sci Prog. 2021;104(2):368504211026409. doi:10.1177/00368504211026409

-Islam MR, Quaiyum S, Pakhe SA, Repon MAU, Bhuiyan MA. Dataset concerning the mental health of healthcare professionals during COVID-19 pandemic in Bangladesh. Data Brief. 2021;39:107506. doi:10.1016/j.dib.2021.107506

Author's response

Thank you very much for this important comment. The introduction part has been extensively revised and similar findings from other similar settings are now provided. With more extensive literature review and discussing other relevant studies, the justification of this study is also changed, as suggested.

Thanks for the suggested articles. We have used them and some other relevant papers in the literature review and discussion (Pages 3-5, lines 61-91; Pages 15-16, lines 256-257, 274-275). Also see the list of references (Pages 21-25).

Reviewer’s comment 

4. The epidemiological data presented in this paper are also very old, they should be updated to let the readers know about the up-to-date information. 

Author's response

The updated epidemiological data up to the date of the initial submission of this manuscript (September 2022) are now provided (Page 5, lines 103-106). 

Reviewer’s comment 

5. Getting 851 responses by inviting 1000 potential respondents are extremely rate in the context of biasness free online survey. As the global response rate is approximately 25% for all online survey. How the authors eliminate bianess of their study. How they selected participants and what was the assumtions befre data collection, these explanation and information should be given in the method section. 

Author's response 

Thanks for bringing up this point. The high response rate could be related to choosing a specifically targeted sample known to have access to the internet and social media. We also made follow up with the invited participants through sending follow up messages and re-sending the survey link. These aspects and the details of selection of participants and the biasness possibilities are now highlighted in the manuscript (Methods: Page 6, lines 117-128; and Limitations: Pages 19-20, lines 356-365). 

Reviewer’s comment 

6. Why the authors took 70% female participants??? Was it randon effect, pls give explanation and potential consequences on the results.

Author's response

Many thanks for this important comment as it is important to provide explanations to this fact in the manuscript. The possible cause of this high number of female respondents is mentioned and the consequences and limitations of this aspect is now highlighted and discussed (Page 20, lines 350-353).

Reviewer #2: 

Reviewer’s comment 

I appreciate the opportunity to review the manuscript entitled "Prevalence and determinants of anxiety and stress in the general population during COVID-19 pandemic”. The study aimed to determine the prevalence of anxiety and stress among the general population during the outbreak of COVID-19 and assess their associated factors. This study is relevant and provides a significant contribution to the knowledge about the psychological impact of the 1st wave of COVID. However, there are some issues that must be addressed prior to publication. I have included my comments and recommendations below: 

Author's response

Thank you very much for the useful comments

Reviewer’s comment 

Introduction

P. 3, ln 61: The authors state: “limited research has addressed the direct effect of COVID-19 on mental health” – nowadays, there are a lot of research on this matter, so this information must be updated. The authors should refer to existing information, as well as the profile of people with more risk factors in terms of their mental health, in order to see if, in the area in question, the profile is similar or different. I think the relevance will be more in this sense, because there are already many studies.

Author's response

Thank you very much for this important comment. The introduction part has been extensively revised and updated literature from other similar settings are now provided. With a more extensive literature review and discussing other relevant studies, the justification of this study is also rephrased, as suggested. New references were also used in the discussion (Pages 3-5, lines 61-91; Pages 15-16, lines 256-257, 274-275). Also see the list of references (Pages 21-25)

Reviewer’s comment 

Method

Study Participants 

Additional information is needed about the key informants and the participants criteria that based the initial selection od the 1000 people. 

Author's response

Additional information about the key informants and the participation criteria are now provided (Page 6, lines 124-128).

Reviewer’s comment 

If the questionnaire was sent through social media, is it possible that people without that kind of access were excluded, biasing the sample? As can be seen in the description of the participants, it is mostly young people who responded, which makes sense considering the way the questionnaire was disseminated. The older people are a minority of the sample, which should be mentioned in the limitations of the study.

Author's response

Thanks for raising this concern. This limitation of using an online questionnaire and sending it through social media is now highlighted (Pages 19-20, lines 354-359).

Reviewer’s comment 

Measures 

Reliability information must be presented, for both scales.

Author's response

The validity and reliability of the questionnaire is now provided (Page 8, lines 161-165).

Reviewer’s comment 

Results

If the authors use scales like the Perceived Stress Scale and the Anxiety Scale, it makes no sense to present the statistics for each item, only the overall stress score.

Author's response

The statistics of each item of the two scales are now removed (old tables 2 and 3 and their texts are removed) (Pages 10-11).

Reviewer’s comment 

The statistical information presented, such as t-test results, should be presented in greater detail, not only the p-values

Author's response

Details of the tests are now provided (Table 3, page 12-13)).

Reviewer’s comment 

Discussion

In the discussion there is no point in re-presenting statistical values.

Author's response

Re-presentation of the statistical values are removed (Page 15, lines 249-250, Page 16, 267).

Reviewer’s comment 

How can the authors guarantee that the values found are due to the pandemic impact and do not reflect pre-pandemic values? 

This should also be mentioned in the limitations, since this is a cross-sectional study

Author's response

This limitation is now discussed (Page 19, lines 346-350).

Reviewer #3: 

Reviewer’s comment 

This is a well-written manuscript that highlights a topical issue of interest to global health. However, the paper can be improved by considering the under-listed comments.

Author's response

Thank you very much for the helpful comments.

Reviewer’s comment 

1. Line 90: What is the total size of the population? How did you arrive at the sample size used?

Author's response

 Information about the total size of the population and sample size calculation are now provided (Page 5 and 6, lines 108-116).

Reviewer’s comment 

2. Line 91: The inclusion and exclusion criteria should be clearly stated.

Author's response

The inclusion and exclusion criteria are now mentioned (Page 6, lines 117-120).

Reviewer’s comment 

3. Lines: 92 to 96: The statements there should be moved to a section on sampling.

Author's response

Many thanks for these important suggestions on better organizing the methods sections. The statement was moved to the new section of sampling (Page 6, lines 122-129).

Reviewer’s comment 

4. Line 96 to 98: From the sentence beginning “The online …. Should be moved to a section on data collection.

Author's response

This sentence is moved to a new section of data collection (Page 8, lines 166-170).

Reviewer’s comment 

5. The authors used an online questionnaire to gather the data. What shows that the intended respondents answered the questionnaire? The limitations of the study should be stated including the weaknesses of using the approach to data collection employed.

Author's response

This limitation is now highlighted (Page 19-20, lines 354-359)).

Reviewer’s comment 

6. The discussion section is highly descriptive. The authors can improve upon it by providing explanations as to why they reported a particular result and also explain why a particular study support or does not support their study.

Author's response

Thanks for this important comment. The discussion section is revised and explanation of the findings and differences with other studies are now provided (Pages 15-18, lines 258-266, 277-281, 301-306, 313-317, 330-332).

Reviewer’s comment 

7. The paper requires minor editing.

Author's response

Extensive English language editing was carried out.

---

## [Decision Letter · Decision Letter 1]

6 Mar 2023

Prevalence and determinants of anxiety and stress in the general population during COVID-19 pandemic in Iraq: A cross-sectional study

PONE-D-22-24670R1

Dear Dr. Shabila,

We’re pleased to inform you that your manuscript has been judged scientifically suitable for publication and will be formally accepted for publication once it meets all outstanding technical requirements.

Kind regards,

Suhad Daher-Nashif, MSc., PhD

Academic Editor

PLOS ONE

Reviewers' comments:

Reviewer's Responses to Questions

**Comments to the Author**

1. If the authors have adequately addressed your comments raised in a previous round of review and you feel that this manuscript is now acceptable for publication, you may indicate that here to bypass the “Comments to the Author” section, enter your conflict of interest statement in the “Confidential to Editor” section, and submit your "Accept" recommendation.

Reviewer #1: All comments have been addressed

Reviewer #2: All comments have been addressed

2. Is the manuscript technically sound, and do the data support the conclusions?

Reviewer #1: Yes

Reviewer #2: (No Response)

3. Has the statistical analysis been performed appropriately and rigorously? 

Reviewer #1: Yes

Reviewer #2: (No Response)

4. Have the authors made all data underlying the findings in their manuscript fully available?

Reviewer #1: Yes

Reviewer #2: (No Response)

5. Is the manuscript presented in an intelligible fashion and written in standard English?

Reviewer #1: Yes

Reviewer #2: (No Response)

6. Review Comments to the Author

Reviewer #1: Thank you for addressing my previous comments and suggestions. I think the manuscript has significantly improved from this revison.

Reviewer #2: (No Response)

7. PLOS authors have the option to publish the peer review history of their article (what does this mean?). If published, this will include your full peer review and any attached files.

Reviewer #1: **Yes: **Md. Rabiul Islam

Reviewer #2: **Yes: **Rute Brites

---

## [Editor Report · Acceptance letter]

4 Apr 2023

PONE-D-22-24670R1 

Prevalence and determinants of anxiety and stress in the general population during COVID-19 pandemic in Iraq: A cross-sectional study 

Dear Dr. Shabila:

I'm pleased to inform you that your manuscript has been deemed suitable for publication in PLOS ONE. Congratulations! Your manuscript is now with our production department. 

Kind regards, 

on behalf of

Dr. Suhad Daher-Nashif 

Academic Editor

PLOS ONE